# Study of the Quality Parameters and the Antioxidant Capacity for the FTIR-Chemometric Differentiation of *Pistacia Vera* Oils

**DOI:** 10.3390/molecules25071614

**Published:** 2020-04-01

**Authors:** Lydia Valasi, Dimitra Arvanitaki, Angeliki Mitropoulou, Maria Georgiadou, Christos S. Pappas

**Affiliations:** 1Laboratory of Chemistry, Department of Food Science & Human Nutrition, Agricultural University of Athens, Iera Odos 75, 11855 Athens, Greece; lydia.valasi@aua.gr (L.V.); dimitrarvntk@gmail.com (D.A.); aggelikimitropoulou8@gmail.com (A.M.); 2Laboratory of Food Process Engineering, Department of Food Science & Human Nutrition, Agricultural University of Athens, Iera Odos 75, 11855 Athens, Greece; m.georgiadou@aua.gr

**Keywords:** *Pistacia vera*, antioxidant, quality, tocopherol, FTIR, discriminant analysis

## Abstract

The aim of this work was to characterize the pistachio oil of the Greek variety, “Aegina”, evaluate its various quality indices, and investigate the potential use of FTIR as a tool to discriminate different oil qualities. For this purpose, the antioxidant capacity, the tocopherol content and the oxidation and degradation of fatty acids, as described by k, Δk, R-values, and free acidity were evaluated using 45 samples from eight different areas of production and two subsequent years of harvesting. The antioxidant capacity was estimated using 2,2′-azinobis(3-ethylbenzothiazoline-6-sulfonic acid diammonium salt (ABTS) and 2,2-diphenyl-1-(2,4,6-trinitrophenyl)hydrazine (DPPH) assays, and the tocopherol content was quantified through HPLC analysis. FTIR spectra were recorded for all samples and multivariate analysis was applied. The results showed significant differences between the oil samples of different harvesting years, which were successfully discriminated by a representative FTIR spectral region based on R-value, total antioxidant capacity, and scavenging capacity, through ABTS. A similar approach could not be confirmed for the other quality parameters, such as the free acidity and the tocopherol content. This research highlighted the possibility of developing a simple, rapid, economic, and environment friendly method for the discrimination of pistachio oils according to their quality profile, through FTIR spectroscopy and multivariate analysis.

## 1. Introduction

*Pistacia*, a genus of the *Anacardiaceae* family, includes at least eleven species, among them *Pistacia vera* L. is the only edible commercial species [1]. The pistachio nut is an important agricultural commodity for a number of countries. Iran, United States, Turkey, Syria, Greece, Italy, and Spain are the main pistachio producers [2]. Pistachio nut can be considered to be a functional food, and has recently been ranked among the first 50 food products with the highest antioxidant potential [3]. The Dietary Guidelines recommend that consuming nuts (almonds, hazelnuts, walnuts, pistachios, pecans, and peanuts) as a part of a daily diet has a beneficial effect on human health [4].

The increasing consumption and demand for novel edible oils has led to a market expansion for the plant-derived oils, which receive particular attention due to their attractive sensory characteristics and their high nutritional properties [5]. The pistachio (*Pistacia vera* L.) oil content ranges from 50% to 60% (dry weight) in kernels, depending mainly on the cultivar, crop year, and geographic location. Even though no specific standards for pistachio oil have been set by the Codex Alimentarius on Fats and Oils, it is claimed to be a niche product [6,7]. It has gained attention due to its special organoleptic characteristics [8] and its richness in some nutrients and health promoting compounds that exhibit high antioxidant capacity [9,10].

The most abundant components in pistachio oil are the fatty acids profiled as (mono- and poly-unsaturated, saturated, and as esters of triglycerides) [5]. The structure of fatty acids might change due to oxidation caused by bad agricultural practices, inappropriate harvesting time, and storage conditions, which can be detected by ultraviolet-visible (UV–Vis) spectroscopic measurements. Absorbencies at 232, 268, 270, and 274 nm are correlated with the state of oxidation, through the detection of secondary oxidation compounds, and possible adulteration with refined oils. Delta (Δ) k and R-value, resulting from the k_232_, k_268_, k_270_, and k_274_ values, are quick indicators of quality assessment of pistachio oil discriminating between high and poor quality oil and are correlated with possible adulteration [11]. Furthermore, the hydrolysis of the oil results in the formation of free fatty acid (FFA) and glycerol residues, indicating that higher quality oils exhibit very low FFA percentage and acidity [12]. As a result, free acidity is claimed to be an early indicator of the potential storage stability of the product. Additionally, it should be mentioned that the formation of oxidized compounds in pistachio oil is blocked by its own natural preservatives that carry out an important antioxidant activity. Pistachio oil contains numerous phenolic compounds that increase its shelf life, and prevent or reduce the damage to cells, caused by free radicals. The evaluation of the total antioxidant capacity is certainly a very useful property of the pistachio oil, ensuring the preservation of the most important health benefits and sensory characteristics. Additionally, tocopherols’ content is associated with health benefits, as confirmed by clinical evidence [13]. Tocopherols are the main antioxidant components that are equally active to vitamin E, thus, are considered to be important biofunctional compounds of the human diet. Due to their non-polar nature, their presence in oils is profound [7]. The major tocopherol isomer in pistachio oil is γ-tocopherol, the most prevalent form of vitamin E [14].

Published works have shown that Fourier transform infrared (FTIR) spectroscopy combined with statistical methods for the discrimination of wines, according to the variety and vintage year, can be used as a rapid, accurate, simple, environment friendly, and economical approach [15,16]. The chemical composition of pistachio oil might be influenced by several factors like variety, climatic and agronomic conditions (weather, soil), the growing season, and agricultural practices [17,18]. Very limited data are available in the literature for the quality profile of the pistachio oil that is extracted from the main Greek pistachio variety, “Aegina”.

The present work focused on the characterization of the pistachio oil of the variety “Aegina”, evaluated various quality indices, and investigated the potential use of FTIR as a tool to discriminate different oil qualities.

## 2. Results and Discussion

All pistachio oil samples were stored in a freezer (−20 °C), in order to maintain their initial quality until analysis. Storage at low temperatures prevents from increasing or reducing the concentrations of oil components and helps to maintain the oil’s primary quality [19].

### 2.1. Oil Extraction

The oil yield ranged between 59.7%–68.2% *w*/*w*, with an average of 62.5 ± 2.6% *w*/*w* for the samples of 2017 and between 52.5%–64.8% *w*/*w* for the samples of 2018, with a mean value of 59.6 ± 2.8% *w*/*w*, respectively. No statistically significant differences were observed in the oil yield between the samples obtained in the two years.

### 2.2. Evaluation of Antioxidant Capacity

Total antioxidant capacity (TAC) and scavenging capacity of the analyzed pistachio oil samples, as measured by the DPPH and ABTS assays are shown in Table 1.

The Trolox calibration curve equations used for transforming absorbance inhibition values (AI) to Trolox equivalents (TE, mM) for the DPPH and ABTS assays, were Equations (1) and (2), respectively.
AI_DPPH_ = (−0.388) × TE + 0.797, R^2^ = 0.971(1)
AI_ABTS_ = (−0.675) × TE + 0.695, R^2^ = 0.995(2)

Antioxidant capacity, as determined by the DPPH assay, ranged between 3.11–6.70 mM with a mean TAC value of 5.03 ± 1.3 mM, and between 0.10–10 with a mean TAC value of 5.57 ± 2.56 mM, for the samples of 2017 and 2018, respectively. Following this, the scavenging capacity ranged between 28.75%–44.29% and 28.96%–52.29% for the samples of 2017 and 2018, with mean values of 36.89 ± 4.33% and 38.62 ± 5.55%, respectively.

TAC values, as determined by the ABTS assay, ranged between 4.92–10.02 mM and 7.07–10.00 mM for the samples of 2017 and 2018, respectively, resulting in mean TAC values of 8.00 ± 1.68 mM and 8.84 ± 0.84 mM. The scavenging capacity, as measured by the ABTS assay, ranged between 42.53%–97.25%, with an average of 76.39 ± 17.75% for the 2017 samples and between 61.75%–96.6% for the 2018 samples, with a mean value of 83.17 ± 9.56%.

Concerning mean TAC and mean scavenging capacity values, results from the ABTS assay were significantly and consistently higher than those from DPPH in both years of harvest. This was due to the applicability of DPPH to hydrophobic systems. Specifically, DPPH was discolored in the presence of compounds that were capable of either transferring an electron or donating hydrogen (lipophilic components). On the other hand, ABTS was freely soluble in both organic and aqueous solvents, thus, it could be used to screen both hydrophilic and lipophilic antioxidants, exhibiting a better estimation of the overall antioxidant capacity of the foods [20,21]. Consequently, the results that were obtained from the ABTS assay were only considered for further statistical analysis. TAC and scavenging capacity, as estimated by ABTS were statistically different between 2017 and 2018. The differences were statistically significant.

### 2.3. UV-Vis Spectroscopic Assessment

The quality indices associated with the k_232_, k_268_, k_270_, k_274_, Δk, and R values were evaluated in 45 pistachio oil samples and the results are displayed in Table 2. The European Quality Standard of Commission Regulation (EEC) No 2568/91 (Annex IX of the Regulation) has set the standard values for extra virgin olive oil (EVOO), as described in Table 2. Considering that an official protocol to predict the quality indicators of the pistachio oil or other nut oils based on the Δk and R-value has not been established, the existing limits were used for the evaluation of pistachio oil samples.

The quality of the oil was assessed by the UV–Vis absorption screening, which identifies changes in the structure of fatty acids due to oxidation. A low absorption in this region is indicative of high-quality oil, whereas old, refined, and generally poor-quality oils show a greater level of absorption in this region, implying high degree of oxidation. The absorbance at 232 nm is caused by hydroperoxides (primary stage of oxidation) and conjugated dienes (intermediate stage of oxidation). The absorbance at 270 nm was caused by carbonylic compounds (secondary stage of oxidation) and conjugated trienes (technological treatments). In the oils, due to oxygen fixation in linolenic and linoleic acids’ double bond position, hydroperoxides arise. The double bond provokes the formation of conjugate diene systems between the carbon atoms. This kind of conjugate systems presents a maximum absorption at 232 nm. During more advanced oxidation states, the products are generated with conjugate diene systems of carbon–oxygen. The maximum absorption in this case ranges between 260–280 nm [11].

The mean k values for each harvesting year were 0.15 ± 0.02 (2017), 0.07 ± 0.01 (2018), 0.01 ± 0.01 (2017), 0.01 ± 0.00 (2018), 0.02 ± 0.01 (2017), 0.01 ± 0.00 (2018), 0.02 ± 0.01 (2017), and 0.01 ± 0.00 (2018) for k_232_, k_268_, k_270_, and k_274_, respectively. Δk was 0.00 ± 0.00 for all samples, regardless of the year of harvest or the origin. The mean R-values were 11.13 ± 3.00 and 5.42 ± 0.91, for the samples of 2017 and 2018, respectively.

There were no significant differences between the years of harvest, based on k and Δk, as all measurements ranged into the high-quality limits. As for the R-value, the 2017 samples were systematically higher than 2018 samples. Specifically, 82% of the samples, which mostly originated from 2018 harvest complied with the EVOO standard, except for the R-value of the remaining 18% of the total samples, which belonged to the 2017 harvest (samples 1, 3, 4, 7, 8, 11, 13, and 14).

### 2.4. Acid (AV) and FFA Values

AV value is a measure of the number of carboxylic acid groups. It is used as an indicator for edibility of oil and is expressed in milligrams per gram. However, FFA are expressed as a percentage of oleic acid. According to the Codex Standard for Edible Fats and Oils, acid value of oil suitable for edible purposes should not exceed 4 mg/g.

FFA has been reported to play a very important role in the aroma and flavor. FFA also contributes to the organoleptic quality of foods, when present in adequate concentration. FFA content is an index of lipase activity and an indicator of freshness, storage time, and stability of many fat-rich foods. It is well-known that FFAs are more susceptible to lipid oxidation, leading to rancidity and production of off-odor, compared to intact fatty acids in triglycerides. It is considered to be an early indicator of the storage stability of the oil [22], with a supreme limit that is less than 0.35% [11].

As presented in Table 2, the AV ranged from 0.53 to 6.61 and from 0.63 to 8.12 mg/g for 2017 and 2018, respectively. The mean AV (1.19 ± 1.32 mg/g oil) of 2018 exhibited lower values than the mean AV (2.12 ± 1.46 mg/g oil) of 2017. The FFA content ranged from 0.27% to 3.33% and from 0.32% to 4.08% for 2017 and 2018, respectively. Similarly, the mean FFA (0.60 ± 0.66%) of 2018 was lower than the mean FFA (1.07 ± 0.73) of 2017, leading to the conclusion that the 2018 harvesting exhibited a superior antioxidant capacity. Based on the standard for edible oils, only two samples (No. 1 and No. 24) from the area of Aegina showed values out of the acceptable levels of AV, whereas only two samples (No. 9 and No. 40) exhibited acceptable levels of FFA content.

### 2.5. Tocopherol Analysis

The limit of detection (LOD) for tocopherol analysis was 0.15 μg/mL. The tocopherol calibration curves used for the qualitative separation of samples were (Equations (3)–(6)):Area (mV x s) = 8.0556 × C_α-T_ (μg/mL) + 0.531, R^2^ = 0.999(3)
Area (mV x s) = 10.724 × C_β-T_ (μg/mL) + 1.9311, R^2^ = 0.998(4)
Area (mV x s) = 13.786 × C_γ-T_ (μg/mL) + 2.5428, R^2^ = 0.978(5)
Area (mV x s) = 14.617 × C_δ-T_ (μg/mL) + 2.1461, R^2^ = 0.997(6)

As for recovery evaluation, the amount of vitamin E isomers added to the samples corresponded to 98.47%, 77.86%, 47.44%, and 110.37% (Equation (14)) of the expected α-T, β-T, γ-T, and δ-T, and the intraday analytical precision was 3.08%, 5.99%, 4.89%, and 2.75% (Equation (15)), respectively.

Figure 1 illustrates the separation of the most important vitamin E isomers, as determined with the HPLC method, using fluorescence detection. The retention times for the α-, β-, γ-, and δ-tocopherols were 8, 10, 12, and 16 min, approximately. The concentration of each tocopherol for all 45 pistachio oil samples is presented in Table 3. The results were obtained and corrected on the basis of recovery and repeatability of the method, as determined by the coefficient of variation (CV). The tocopherol contents of pistachio oils expressed as 10^2^ μg/mL pistachio oil, ranged from 0.53 (No. 25) to 5.90 (No. 43), 0.33 (No. 13) to 2.25 (No. 29), 97.56 (No. 36) to 235.06 (No. 6), and 0.84 (No. 12) to 2.31 (No. 20) for α-, β-, γ-, and δ-tocopherol, respectively. The above minimum and maximum values corresponded to 13.25–147.50, 8.25–56.25, 2439.00–5876.50, and 21.00–57.75 mg/kg of pistachio oil for α-, β-, γ-, and δ-tocopherol, respectively. It is important to mention that γ-tocopherol is coeluted with β-tocotrienol, as a result, the calculated content of γ-tocopherol includes both isomers. The data indicate that the main form in all samples was γ-tocopherol (by coelution with β-tocotrienol), whereas the β-tocopherol content was limited. These results are in agreement with Martinez et al. (2016) [23]. The minimum and maximum values of each tocopherol presented in the samples were compared to the standard for vegetable oils provided by the Codex Alimentarius Commission on Fats and Oils (Table 4). With regards to pistachio oils, the quantity of α- and δ-tοcopherol, as measured in the present work for the variety “Aegina”, ranges within the limits that have been set by the standard. However, no values were described for the β-tocopherol and β-tocotrienol, in contrast to the present study.

The mean values of each harvesting year were 1.74 ± 0.72 (2017), 2.60 ± 1.15 (2018), 0.77 ± 0.31 (2017), 1.19 ± 0.55 (2018), 192.81 ± 26.41 (2017), 178.12 ± 32.01 (2018), 1.55 ± 0.33 (2017), and 1.80 ± 0.29 (2018), expressed as 10^2^ μg/mL pistachio oil for α-, β-, γ-, and δ-tocopherol, respectively. The aforementioned values corresponded to a total tocopherol content of 196.87 for the 2017 harvest and 183.71 for 2018 (expressed as 10^2^ μg/mL pistachio oil).

### 2.6. FTIR Spectroscopy Study

Figure 2 shows two representative FTIR spectra of a pistachio oil sample with its basic peaks marked. The presented spectra depict samples of common origin but of different harvesting year. It is interesting to note that both spectra are optically very similar and, thus, the use of discriminant analysis is necessary. Each peak corresponds to a certain wavenumber that is attributed to specific vibrations and chemical structures of components from pistachio oil (Table 5).

Pistachios are rich in lipids (48%–63%), with a balanced content of mono- (56%–77%) and poly-unsaturated (10%–31%) fatty acids, protein (18%–22%), and dietary fibers (8%–12%). Moreover, they present a high content of bioactive compounds, such as tocopherols, phytosterols, and phenolic compounds [32]. Main lipid acids absorb in the same spectral region (3007–772 cm^−1^) as phenols, tocopherols, and sterols [24,26,28,33,34].

### 2.7. Statistical Analysis

In case the number of samples for each test group exceeded 30, Levene’s test (*t*-test) was applied without testing whether the data were normally distributed or not. Additionally, a normality test was applied in order to accept or reject the null hypothesis that each test group was statistically different from a normal distribution. Kolmogorov–Smirnov’s and Shapiro–Wilk’s normality tests evaluated if the groups followed normal distribution (*p*-value > 0.05), with the Shapiro–Wilk’s result exhibiting higher validity, as it comes from a more conservative test. If the data did not follow a normal distribution, the appropriate normalization was made to fix the skewness and kurtosis values, at the accepted levels.

When the normality test was confirmed, Levene’s test was used to assess the equality of variances. Levene’s test checked the null hypothesis that the test group variances were equal (homogeneity of variance or homoscedasticity). If the resulting *p*-value of Levene’s test was less than the required significance level (typically 0.05), the obtained differences in sample variances were unlikely to have occurred, based on random sampling from a population with equal variances, so the test group were significantly different.

MetaboAnalyst checked data integrity and continued on to data filtering. The purpose of the data filtering was to identify and remove variables that were unlikely to be of use when modeling the data. This step is strongly recommended for datasets with a large number of variables, many of which are from baseline noises. Based on the total number of variables, 10% of data were filtered, logarithmically transformed, and auto-scaled (mean-centered and divided by the SD of each variable).

The total number of samples (45 pistachio oil samples) was differentiated according to their year of harvest.

#### 2.7.1. Discrimination Based on Antioxidant Capacity

Levene’s test of SPSS was used to assess the equality of ABTS variances for two years of harvest, 2017 and 2018. Levene’s test tested the null hypothesis that the ABTS variances of 2017 and 2018 were equal. *P*-value less than 0.05 rejected the null hypothesis and proved that the TAC (Wilks’ Lambda = 0.895) and the scavenging capacity (Wilks’ Lambda = 0.939) of ABTS were different in 2017 and 2018. From cross-validated grouped cases, 71.10% were classified correctly according to their antioxidant capacity and year of harvest.

#### 2.7.2. Discrimination Based on R-Value Study

Based on R-values, 86.70% of cross-validated grouped cases were correctly classified to their year of harvest and the *P*-value (<0.001) proved the accuracy and robustness of the forecasting model, using SPSS. Therefore, the R-value of the samples was exploited to classify the samples according to the year of production (Wilks’ Lambda = 0.315).

#### 2.7.3. Discrimination Based on Acid Value and Free Fatty Acid

Levene’s test examined the null hypothesis that the AV of 2017 and 2018 harvest were equal and the same assumption was made for the FFA content. The results (*p*-value > 0.05) failed to reject the null hypothesis and indicated that AV and FFA were not significantly different between the two years of harvest. Discrimination analysis displayed a percentage of correct classification at 71.10% (cross-validated grouped cases).

#### 2.7.4. Discrimination Based on the Tocopherol Analysis

SPSS could not discriminate between the years of harvest of the 45 pistachio oil samples, according to their total tocopherol content. Levene’s test for equality of variances between the 2017 and 2018 harvest exhibited a *p*-value > 0.05 and a 61.40% cross-validation level.

#### 2.7.5. Discrimination Based on FTIR Spectroscopy Study

The spectral regions 3030–2795 and 1805–650 cm^−1^ were selected for the discriminant analysis, i.e., the regions where the peaks were observed (Figure 2). Applying the principal component analysis, the initial set of variables was reduced to a number of hidden variables of principal components (PC). The scree plot (Figure 3) revealed that the greatest impact on the variance of the analysed spectra for the pistachio oil samples was related to the first two principal components. Figure 4 and Figure 5 present the score and loading plot for the principal components (PC) in the principal components analysis (PCA) model. The pistachio oil samples were clearly classified into two groups (2017 and 2018 harvest year). As depicted in Figure 6, MetaboAnalyst could correctly classify 100% of the cross-validated grouped cases, according to their chemical composition and year of harvest with R^2^ = 0.992 and Q^2^ = 0.987, which indicate a high predictive accuracy. *P*-values less than 0.05 proved that the FTIR method could be used as an accurate rapid screening tool for the differentiation of pistachio oils by their year of harvest.

#### 2.7.6. Statistical Models Comparison

The evaluation of the total antioxidant capacity of pistachio oil samples through an ABTS assay showed that TAC and their scavenging capacity could be statistically differentiated, among the years of harvesting, as was also observed with the results based on R-value. It is worth noticing that in the case of AV, FFA, and HPLC-fluorescence analysis, there were no statistically significant differences between 2017 and 2018, at a 95% confidence level. However, FTIR spectroscopy combined with the statistical methods represent an appropriate rapid technique to discriminate pistachio oils of different quality, based on their antioxidant profile.

## 3. Materials and Methods

### 3.1. Samples

A total of 45 pistachio samples of the Greek variety ‘Aegina’ were provided by pistachio farmers from eight different regions of Greece (Aegina, Megara, Phthiotis, Evia, Volos, Trizina, Thiva, and Avlona) during the 2017 and 2018 harvest seasons. Due to alternate bearing, the number of samples of 2017 was less than the succeeding year. The pistachios were sound and had the typical characteristics of the variety. They were dried under the sun or mechanically at moisture level 5%–7%, after dehulling at farm level. In the laboratory, each sample was shelled and finely ground in an IKA M 20 (IKA, Königswinter, Germany) laboratory mill, at a maximum rotational speed 20,000 rpm, followed by particle size separation using sieves (500 µm < size < 800 µm). After preparation, all samples were put in sealed bags, protected from light, and stored in the freezer (−20 °C) until analysis.

### 3.2. Reagents

Petroleum ether, 2,2-diphenyl-1-(2,4,6-trinitrophenyl)hydrazine (DPPH), 2,2′-azinobis(3-ethylbenzothiazoline-6-sulfonic acid diammonium salt (ABTS), potassium persulfate (K_2_S_2_O_8_), potassium hydrogen phthalate (KHP), sodium hydroxide pellets (NaOH), 6-hydroxy-2,5,7,8-tetramethylchroman-2-carboxylic acid (Trolox), ethyl acetate, tetrahydrofuran (THF), n-heptane, cyclohexane 99.8%, methanol (MeOH), and ethanol (EtOH) were purchased from Sigma-Aldrich (Steinhein, Germany). (+)-α-, (+)-β-, (+)-γ-, and (+)-δ-tocopherol standards of 99.99% purity were obtained from Merck (Darmstadt, Germany). Distilled water and phenolphthalein indicator solution were also used. All compounds and solvents were of analytical grade.

### 3.3. Oil Extraction

Pistachio oil was extracted from 4 g of kernel flour with 250 mL of petroleum ether in a Soxhlet apparatus for 6 h, according to the AOAC Official Method 948.22. After evaporation of the solvent under reduced pressure, the oil was weighed to measure the lipids’ mass and was kept in a freezer (–20 °C) to maintain its initial quality, until analysis. The extraction was carried out in triplicates and the mean value with the standard deviation was calculated.

### 3.4. Evaluation of the Antioxidant Capacity

#### 3.4.1. DPPH Assay

DPPH radical-scavenging capacity was determined according to Minioti and Georgiou (2010) [17], with some modifications using a JASCO V-550 spectrophotometer (JASCO Corporation, Tokyo, Japan). Briefly, 100 μL of pistachio oil were mixed with 4 mL of DPPH working solution (8.1 × 10^−5^ M working solution of the DPPH radical in ethyl acetate). The reaction mixture was vigorously stirred for a few seconds and kept in a dark place for 30 min, at room temperature. Absorbencies were measured at 515 nm against a blank (100 μL of ethyl acetate instead of pistachio oil). Pistachio oil antioxidants scavenged the DPPH radical, resulting in decolorization of its purple solution. Analyses were performed in triplicates. The scavenging capacity was calculated using Equation (7):Scavenging capacity = [(A_515_ of control − A_515_ of sample)/A_515_ of control] × 100(7)
A calibration curve (0.08−1 mM) was constructed using Trolox as the external standard and the obtained values were expressed as mmol/L of Trolox equivalents per mL of oil.

#### 3.4.2. ABTS Assay

The ABTS assay was slightly modified, based on the methods of Rajaei et al. (2010) [35] and Torres-Martinez et al. (2017) [36], using an Agilent 8453 spectrophotometer. In brief, 96 mg of ABTS with distilled water were diluted in a 25 mL volumetric flask and 440 μL of K_2_O_8_S_2_ solution (0.14 M in distilled water) were added. The mixture was maintained for 18 h, protected by light, at room temperature for stabilization of the ABTS oxidation. Prior to further use, the ABTS^+^ solution was diluted with EtOH, at an absorbance value of 0.7 ± 0.005 (working solution). Antioxidant capacity was evaluated by measuring the scavenging effect of 100 μL of pistachio oil, mixed with 2 mL of ABTS^+^ working solution, followed by shaking and incubation in the dark, for 6 min at room temperature. The decrease in absorbance was then measured at 734 nm against a control solution (100 μL of EtOH). All measurements were performed in triplicates. The scavenging capacity was calculated using Equation (8):Scavenging capacity = [(A_734_ of control − A_734_ of sample)/A_734_ of control] × 100(8)
Trolox was used as a reference compound for the calibration curve with a concentration range of 0.05–1 mM and a total antioxidant capacity, expressed as mmol/L of Trolox equivalents per mL of oil, was calculated and reported as mean ± SD.

### 3.5. Quality Assessment of Pistachio Oil

#### 3.5.1. UV–Vis Assessment

The Agilent Cary 60 UV–Vis spectrophotometer (Agilent Technologies, Mississauga, ON, Canada) and rectangular quartz cuvettes with an optical length of 1 cm were used according to EEC No 2568/91 (Annex IX of the Regulation). Pistachio oil samples (45 in total) were diluted in cyclohexane. A total of 0.1 g of pistachio oil was weighed accurately into a 10 mL graduated flask, filled up to the mark with the solvent, and homogenized. The resulting solution (10 g/L) was perfectly clear. If opalescence or turbidity was present, it was filtered through the paper. All samples were measured in cuvettes, running a solvent blank as a reference. Absorption measurements for purity determination were made at 232, 268, 270, and 274 nm in triplicates, and the average was used for the determination of pistachio oil purity. K values were calculated according to Equation (9):k = A/(C × s)(9)
where A is the absorbance at the specified nanometer; C is the concentration in grams per liter; and s is the cuvette thickness in centimeter. Delta (Δ) k and R-value were evaluated using Equations (10) and (11):Delta (Δ) k = k_270_ − [(k_268_ + k_274_)/2](10)
R-value was calculated = k_232_/k_270_(11)

#### 3.5.2. Determination of AV and FFA

The AV and FFA content were determined in triplicates, according to Otemuyiwa and Adewusi (2013) [22]. In brief, titration of pistachio oil (1 g) dissolved in 5 mL EtOH was applied, using a 0.1 M NaOH solution as the standard reagent to a phenolphthalein endpoint (when the addition of a single drop of alkali produces a slight but definite color change that persists for at least 15 s). The AV value was expressed as oleic acid, according to Equation (12). All determinations were performed in triplicates. The acid value was calculated according to Equation (13):AV = (56 × C × V)/m(12)
where V is the titration volume (mL) of the standard volumetric NaOH solution used; C is the concentration (M) of the standard volumetric NaOH solution used; and m is the mass (g) of the pistachio oil sample. The percentage of FFAs in the pistachio oil was calculated using Equation (13):% FFA = 0.503 × AV(13)
Then, the AV and FFA mean values and the corresponding SDs were calculated.

### 3.6. Tocopherol Analysis

#### 3.6.1. Apparatus and Chromatographic Conditions

The chromatographic analysis was carried out in an analytical HPLC unit equipped with a JASCO PU 980 pump, with a 100 μL injection loop, a JASCO FP920 fluorescence detector (Co. Ltd., Tokyo, Japan) supported by Clarity Lite software (DataApex, Prague, Czech Republic) for data processing, and an ODS Hypersyl column (4.6 × 250 mm, 5 μm particle size, Thermo Fisher Scientific Inc., Waltham, MA, USA).

The determination of the α-, β-, γ-, and δ-tocopherol (T) content using HPLC, followed the ISO 9936:2006 standard. The mobile phase consisted of the THF/n-heptane (4:96 *v*/*v*) at a flow rate of 1.0 mL/min and the injection volume was 10 μL. The effluent was detected in a fluorescence detector, with an excitation filter at 295 nm and an emission wavelength at 330 nm. The system was operated at ambient temperatures. The tocopherol compounds were identified by chromatographic comparisons of the retention times of the analytes in a standard solution and quantified by the respective calibration curves. The results were obtained from triple measurements, and the mean values and corresponding SDs were calculated.

#### 3.6.2. Standard Solutions

Stock standard solutions, α-T (96.53 μg/mL), β-T (85.74 μg/mL), γ-T (87.40 μg/mL), and δ-T (81.14 μg/mL) in n-heptane were prepared and stored in the dark at −20 °C. Combined working standard mixtures, with concentrations in the expected sample ranges, were prepared daily from the stock standard solutions, by diluting appropriate volumes of stock solutions with n-heptane. Then, a calibration curve for each tocopherol was constructed.

#### 3.6.3. Validation Method

Calibration and linearity: Calibration curves were prepared using standard solution of vitamin E isomers at nine concentrations (C), ranging from 0.15–20 μg/mL.

Recovery: Extraction recoveries were evaluated by adding known amounts of isomers (+)-α-T, (+)-β-Τ, (+)-γ-Τ, and (+)-δ-Τ to the pistachio oil samples. The amounts added were of low, medium and high tocopherol content (0.2, 10 and 20 μg/mL). Recovery was calculated by Equation (14):Recovery = (C of spiked sample/(C of sample + C of standard added)) × 100(14)

Analytical precision: Interday precision was determined by analyzing two concentrations (15 and 20 μg/mL) of standard vitamin E isomers in three replicates on three different days. The following equation was used:Precision, % = (SD/Mean C) × 100(15)

The repeatability of the tocopherols’ measurements in pistachio oil was calculated by Equation (15).

### 3.7. FTIR Spectra Recording

The FTIR spectra of the pistachio oil samples were recorded in triplicates on a Thermo Nicolet 6700 FTIR spectrophotometer (Thermo Electron Corporation, Madison, WI, USA) equipped with a deuterated triglycine sulfate (DTGS) detector. The spectra were in an attenuated total reflection (ATR) mode with a Horizontal ATR accessory from Spectra-Tech Inc. (Stamford, CT, USA). The accessory was equipped with a ZnSe-ATR crystal of a trapezoid shape (800 × 10 × 4 mm). The crystal provided an angle of incidence of 45° and was enclosed in a stainless-steel cuvette. For spectra recording, an aliquot of 200 μL of pistachio oil or tocopherol standard mixture was poured on the ATR crystal and allowed to dry, forming a uniform film. Spectra were recorded with a resolution of 4 cm^−1^ and 100 scans. The speed of the interferometer moving mirror was 0.6329 mm/s. Background spectrum was collected using only ATR crystal, prior to spectrum recording of each sample.

FTIR spectra were smoothed using the Savitsky–Golay algorithm (5-point moving second-degree polynomial) and the baseline was corrected using the ‘automatic baseline correct function’ (second-degree polynomial, twenty iterations). Then, the average spectrum of each sample was measured and normalized (absorbance maximum value of 1). Each average spectrum was extracted and saved as a csv file for their use in discriminant analysis. Spectral data collection and processing was carried out using the OMNIC ver. 8.2.0.387 software (Thermo Fisher Scientific Inc., Waltham, MA, USA).

### 3.8. Statistical Analysis

Discriminant analysis was performed using IBM SPSS Statistics 22 (ver. 8.0.0.245) (SPSS Inc., Chicago, IL, USA) and the MetaboAnalyst 4.0 software (McGill University, Montreal, QC, Canada) for a comprehensive and integrative data analysis.

## 4. Conclusions

The results of this work showed that the pistachio oil samples of the variety “Aegina” were within the limits set by the specific standards in terms of high quality. The oil yields of the samples from the two harvest seasons (2017, 2018) were found to be similar, while statistically significant differences were evident for the antioxidant capacity and the R-value between pistachio oil samples, from different years of harvesting. These differences might be attributed to agroclimatic factors, such as different agricultural practices, average temperature, and rainfall from year to year. The FTIR spectroscopy succeeded to classify pistachio oil samples according to the differences which are related to quality parameters, particularly described by the antioxidant capacity, and the R-value. The developed method is fast, accurate, non-destructive, with no excessive sample preparation, and has the additional advantage of not requiring the use of large quantities of solvents, being especially suitable for the screening of large number of samples. Furthermore, the present results provide evidence that the FTIR method could be a promising discriminating tool against fraud related to plant-derived oils, through the use of quality parameters as indicators.

## Figures and Tables

**Figure 1 molecules-25-01614-f001:**
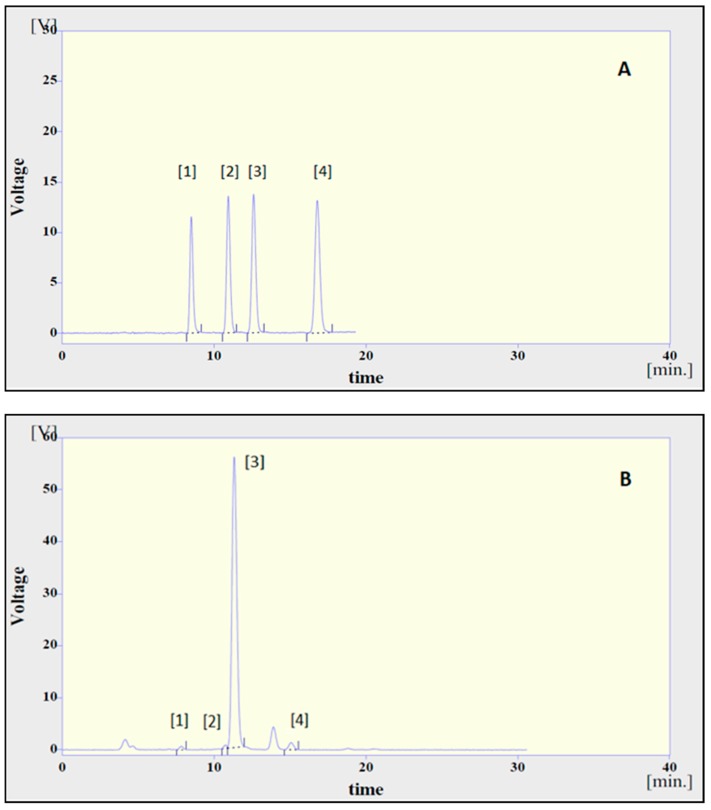
Chromatogram of a working standard mixture (**A**) and of a pistachio oil sample (**B**), determined through high performance liquid chromatography (HPLC)-fluorescence. Peaks: 1, α-tocopherol; 2, β-tocopherol; 3, γ-tocopherol, and β-tocotrienol; 4, δ-tocopherol.

**Figure 2 molecules-25-01614-f002:**
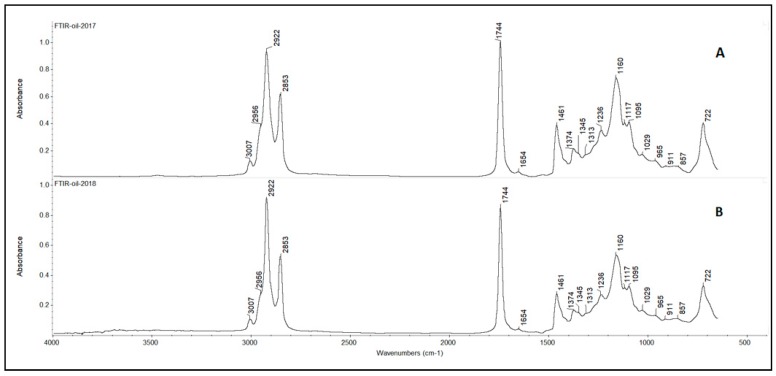
Representative FTIR spectra of pistachio oil samples from the same origin, but from different years of harvest, 2017 (**A**) and 2018 (**B**).

**Figure 3 molecules-25-01614-f003:**
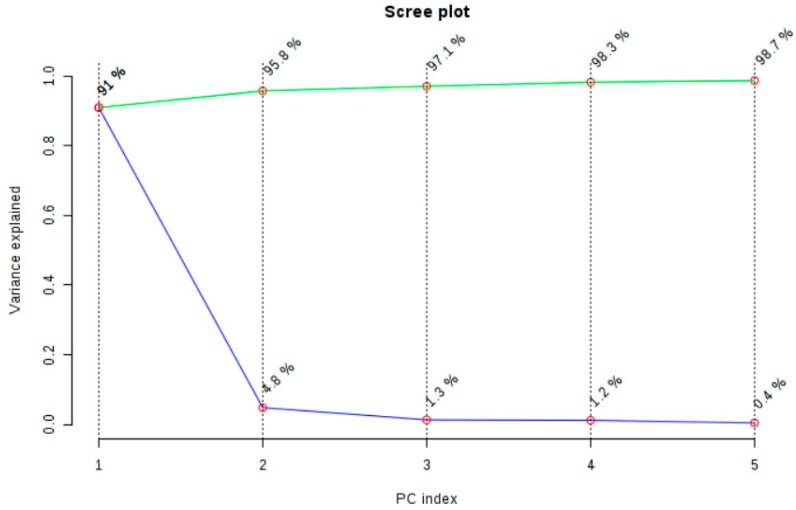
Plot of explained variance for principal components analysis (PCA) of the FTIR spectra.

**Figure 4 molecules-25-01614-f004:**
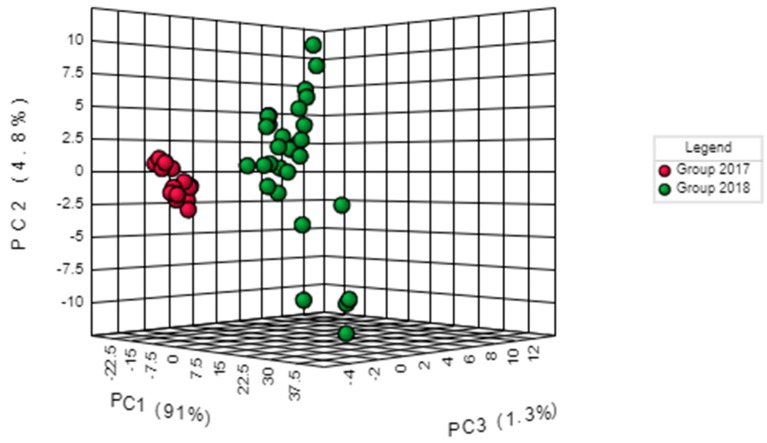
3D Score plot of principal components analysis (PCA).

**Figure 5 molecules-25-01614-f005:**
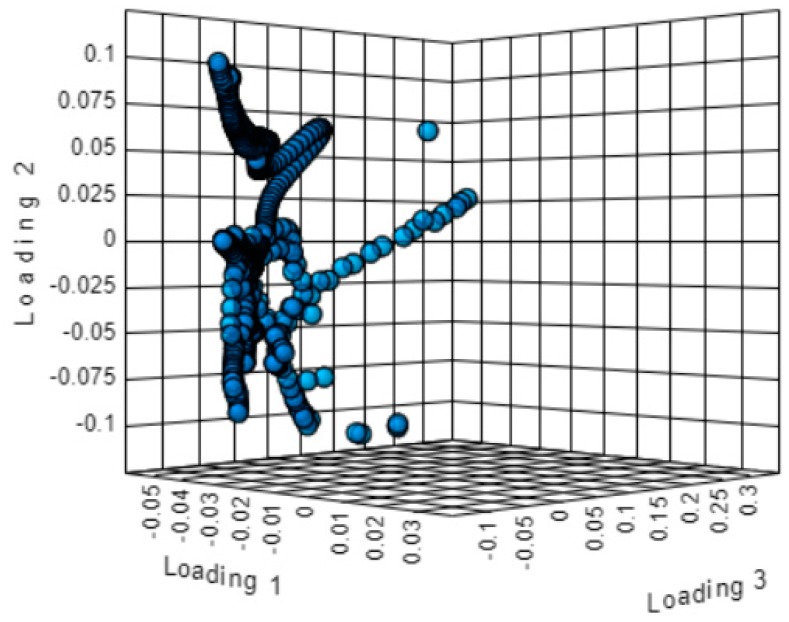
3D loading plot of principal components analysis (PCA).

**Figure 6 molecules-25-01614-f006:**
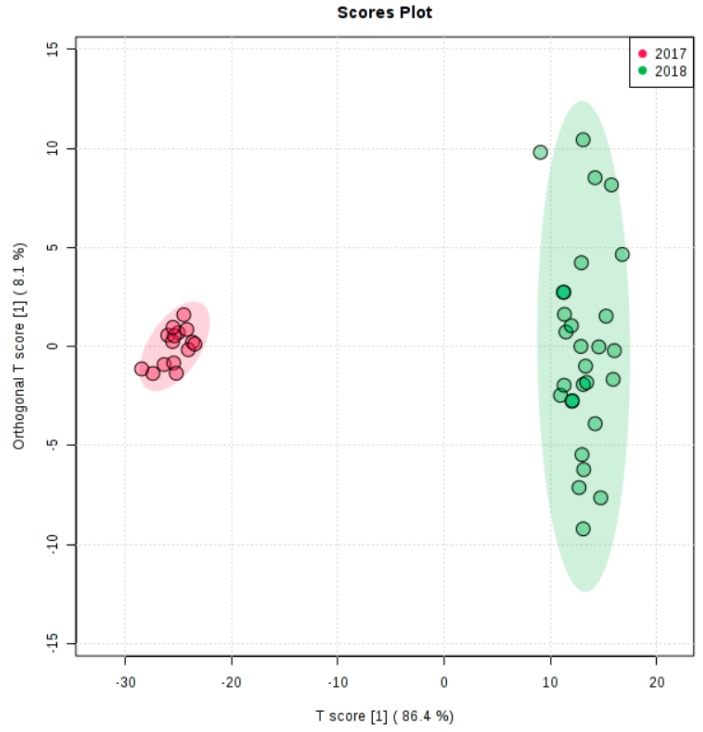
Orthogonal Partial Least Squares–Discrimination (orthoPLS–DA) using MetaboAnalyst.

**Table 1 molecules-25-01614-t001:** Total antioxidant capacity (mM Trolox equivalents per mL pistachio oil) (TAC) and scavenging capacity (%) of pistachio oils of different regions, as determined through the 2,2-diphenyl-1-(2,4,6-trinitrophenyl)hydrazine (DPPH) and 2,2′-azinobis(3-ethylbenzothiazoline-6-sulfonic acid diammonium salt (ABTS) assays.

Samples No	Origin	Year of Harvest	TAC		Scavenging Capacity	
			DPPH	ABTS	DPPH	ABTS
1	AEGINA	2017	6.05	7.10	37.69	68.14
2	AEGINA	2017	6.45	8.63	41.76	82.85
3	AEGINA	2017	6.35	8.60	44.29	84.95
4	AEGINA	2017	3.11	9.45	32.17	91.79
5	AEGINA	2017	3.17	4.92	32.44	45.28
6	AEGINA	2017	5.62	7.44	41.44	74.74
7	AEGINA	2017	3.82	5.46	34.94	48.50
8	MEGARA	2017	5.95	9.32	39.68	90.03
9	MEGARA	2017	5.36	9.01	37.82	85.15
10	MEGARA	2017	3.21	9.44	28.75	90.84
11	MEGARA	2017	6.70	5.02	39.02	42.53
12	PHTHIOTIS	2017	3.59	10.02	31.51	97.25
13	PHTHIOTIS	2017	6.45	9.02	39.42	88.70
14	PHTHIOTIS	2017	4.96	7.85	37.07	73.33
15	TRIZINA	2017	4.68	8.73	35.33	81.85
16	AEGINA	2018	2.12	9.63	39.30	82.26
17	AEGINA	2018	7.92	8.23	37.97	76.33
18	AEGINA	2018	7.10	8.91	42.70	83.65
19	AEGINA	2018	6.49	8.58	40.40	84.02
20	AEGINA	2018	1.99	7.99	40.52	74.89
21	AEGINA	2018	8.44	8.61	47.87	82.05
22	AEGINA	2018	4.83	8.38	34.05	77.05
23	AEGINA	2018	5.86	9.46	40.19	90.34
24	AEGINA	2018	5.30	7.71	34.84	66.41
25	AEGINA	2018	7.47	8.27	38.43	75.91
26	AEGINA	2018	6.70	8.72	41.40	86.10
27	AEGINA	2018	5.69	7.96	36.21	73.62
28	EVIA	2018	5.51	7.07	36.83	64.38
29	EVIA	2018	3.80	9.49	28.96	91.06
30	EVIA	2018	6.67	9.22	40.32	88.08
31	MEGARA	2018	0.28	9.61	32.01	91.97
32	MEGARA	2018	6.47	8.58	24.12	81.59
33	MEGARA	2018	7.35	8.98	38.53	85.76
34	MEGARA	2018	5.34	8.76	35.45	80.68
35	MEGARA	2018	7.45	9.63	43.20	94.15
36	TRIZINA	2018	7.65	8.47	44.10	77.16
37	PHTHIOTIS	2018	5.84	9.80	38.83	93.99
38	PHTHIOTIS	2018	6.04	6.78	40.30	61.75
39	PHTHIOTIS	2018	0.10	9.22	31.87	87.01
40	PHTHIOTIS	2018	1.35	9.83	36.60	94.10
41	PHTHIOTIS	2018	1.40	8.28	36.50	74.18
42	PHTHIOTIS	2018	5.82	10.00	36.09	96.60
43	VOLOS	2018	10.28	9.95	52.29	96.31
44	THIVA	2018	7.34	9.87	43.04	95.39
45	AVLONAS	2018	8.43	9.16	45.60	88.19

**Table 2 molecules-25-01614-t002:** Ultraviolet–visible (UV–Vis) spectroscopy, acid values (AV), and free fatty acid (FFA) of pistachio oil samples against the extra virgin olive oil (EVOO) corresponding values.

Samples No	k_232_	k_268_	k_270_	k_274_	Δk	R	AV ^1^ (as oleic acid)	% FFA ^1^
EVOO	≤2.50	≤0.22	≤0.22	≤0.22	≤0.01	≤11.36	≤4.000	≤0.350
1	0.126	0.009	0.009	0.010	0.000	14.073	6.615 ± 0.000	3.327 ± 0.000
2	0.129	0.015	0.015	0.015	0.000	8.547	3.186 ± 0.325	1.603 ± 0.163
3	0.146	0.010	0.011	0.011	0.000	13.759	2.249 ± 0.000	1.131 ± 0.000
4	0.148	0.011	0.011	0.011	0.000	13.550	1.676 ± 0.019	0.843 ± 0.010
5	0.145	0.017	0.018	0.018	0.000	8.155	1.676 ± 0.019	0.843 ± 0.010
6	0.172	0.020	0.020	0.021	0.000	8.430	1.676 ± 0.019	0.843 ± 0.010
7	0.164	0.010	0.011	0.011	0.000	15.537	3.373 ± 0.000	1.697 ± 0.000
8	0.156	0.013	0.013	0.014	0.000	11.672	1.687 ± 0.000	0.848 ± 0.000
9	0.166	0.020	0.020	0.021	0.000	8.308	0.532 ± 0.001	0.268 ± 0.000
10	0.132	0.014	0.014	0.015	0.000	9.277	1.102 ± 0.000	0.555 ± 0.000
11	0.173	0.027	0.027	0.027	0.000	12.281	2.509 ± 0.336	1.262 ± 0.169
12	0.159	0.018	0.019	0.019	0.000	8.550	1.687 ± 0.000	0.848 ± 0.000
13	0.142	0.009	0.009	0.009	0.000	15.987	1.124 ± 0.000	0.566 ± 0.000
14	0.157	0.013	0.013	0.013	0.000	11.996	1.102 ± 0.000	0.555 ± 0.000
15	0.173	0.025	0.025	0.025	0.000	6.854	1.676 ± 0.019	0.843 ± 0.010
16	0.070	0.011	0.011	0.012	0.000	6.190	0.821 ± 0.007	0.413 ± 0.003
17	0.075	0.016	0.016	0.016	0.000	4.699	0.805 ± 0.269	0.405 ± 0.135
18	0.069	0.011	0.011	0.011	0.000	6.248	1.171 ± 0.166	0.589 ± 0.083
19	0.071	0.011	0.011	0.012	0.000	6.179	0.994 ± 0.161	0.500 ± 0.081
20	0.052	0.007	0.008	0.008	0.000	6.806	0.982 ± 0.164	0.494 ± 0.082
21	0.068	0.011	0.012	0.012	0.000	5.842	1.087 ± 0.009	0.547 ± 0.004
22	0.056	0.012	0.012	0.013	0.000	4.524	1.362 ± 0.270	0.685 ± 0.136
23	0.073	0.014	0.014	0.014	0.000	5.254	0.989 ± 0.148	0.498 ± 0.074
24	0.059	0.020	0.020	0.020	0.000	2.879	8.116 ± 0.191	4.082 ± 0.096
25	0.071	0.011	0.011	0.012	0.000	6.202	0.807 ± 0.014	0.406 ± 0.007
26	0.053	0.008	0.008	0.008	0.000	6.933	1.279 ± 0.163	0.643 ± 0.082
27	0.074	0.015	0.015	0.016	0.000	4.831	0.995 ± 0.166	0.500 ± 0.083
28	0.068	0.012	0.012	0.012	0.000	5.647	1.079 ± 0.015	0.543 ± 0.008
29	0.070	0.013	0.014	0.014	0.000	5.197	0.800 ± 0.251	0.402 ± 0.126
30	0.075	0.019	0.019	0.018	0.000	4.067	0.709 ± 0.614	0.357 ± 0.309
31	0.073	0.015	0.015	0.016	0.000	4.865	1.004 ± 0.156	0.505 ± 0.078
32	0.070	0.011	0.011	0.011	0.000	6.431	1.078 ± 0.023	0.542 ± 0.011
33	0.054	0.011	0.012	0.012	0.000	4.666	0.978 ± 0.151	0.492 ± 0.076
34	0.069	0.014	0.014	0.014	0.000	4.871	0.819 ± 0.277	0.412 ± 0.139
35	0.072	0.014	0.014	0.015	0.000	5.050	1.007 ± 0.157	0.506 ± 0.079
36	0.072	0.017	0.017	0.018	0.000	4.228	0.982 ± 0.161	0.494 ± 0.081
37	0.070	0.012	0.012	0.013	0.000	5.764	0.898 ± 0.151	0.452 ± 0.076
38	0.054	0.009	0.009	0.009	0.000	6.118	1.004 ± 0.157	0.505 ± 0.079
39	0.053	0.010	0.011	0.011	0.000	5.009	0.808 ± 0.016	0.406 ± 0.008
40	0.072	0.013	0.014	0.014	0.000	5.316	0.632 ± 0.155	0.318 ± 0.078
41	0.053	0.011	0.011	0.011	0.000	4.969	1.009 ± 0.160	0.507 ± 0.080
42	0.068	0.012	0.012	0.012	0.000	5.690	0.913 ± 0.164	0.459 ± 0.083
43	0.070	0.012	0.012	0.013	0.000	5.771	0.821 ± 0.004	0.413 ± 0.002
44	0.070	0.010	0.010	0.011	0.000	6.848	0.821 ± 0.007	0.413 ± 0.003
45	0.053	0.010	0.010	0.010	0.000	5.476	0.911 ± 0.166	0.458 ± 0.083

^1^ mean ± SD (*n* = 3).

**Table 3 molecules-25-01614-t003:** Tocopherol (T) content (×10^2^ μg/mL pistachio oil) obtained by Soxhlet extraction and repeatability assessment.

Samples No	Concentration ^1^				Repeatability (CV %, *n* = 3)			
	α-Τ	β-Τ	γ-Τ ^2^	δ-Τ	α-Τ	β-Τ	γ-Τ	δ-Τ
1	1.57 ± 0.16	1.07 ± 0.24	186.77 ± 3.65	1.79 ± 0.04	10.52	22.26	1.95	2.13
2	1.32 ± 0.10	1.14 ± 0.19	202.72 ± 12.77	1.73 ± 0.13	7.45	16.60	6.30	7.35
3	0.99 ± 0.05	0.63 ± 0.09	191.13 ± 4.38	1.57 ± 0.02	4.94	13.88	2.29	1.62
4	1.73 ± 0.15	0.39 ± 0.05	222.09 ± 10.95	1.59 ± 0.05	8.54	12.38	4.93	3.09
5	1.86 ± 0.30	0.45 ± 0.24	187.09 ± 3.09	1.80 ± 0.15	16.21	52.60	1.65	8.10
6	3.12 ± 0.29	1.13 ± 0.46	235.06 ± 7.77	2.11 ± 0.07	9.27	40.99	3.31	3.42
7	2.13 ± 0.12	1.10 ± 0.10	210.49 ± 18.33	1.77 ± 0.14	5.86	8.88	8.71	7.83
8	1.53 ± 0.16	1.06 ± 0.04	174.01 ± 6.78	1.37 ± 0.09	10.31	4.14	3.89	6.62
9	0.59 ± 0.17	0.97 ± 0.17	173.56 ± 11.78	1.41 ± 0.11	29.47	17.01	6.79	8.17
10	1.24 ± 0.07	0.38 ± 0.03	173.29 ± 5.46	1.06 ± 0.09	5.72	7.18	3.15	8.38
11	1.71 ± 0.27	0.98 ± 0.12	170.07 ± 12.59	1.78	15.60	12.51	7.41	0.10
12	1.32 ± 0.18	0.83 ± 0.31	134.59 ± 4.15	0.84 ± 0.22	13.46	37.48	3.08	26.52
13	1.80 ± 0.13	0.33 ± 0.20	199.49 ± 7.96	1.60 ± 0.06	7.49	60.45	3.99	3.48
14	3.37 ± 0.44	0.55 ± 0.15	231.93 ± 9.84	1.17 ± 0.19	12.96	26.87	4.24	16.08
15	1.87 ± 0.17	0.48 ± 0.07	199.88 ± 6.87	1.58 ± 0.12	9.16	14.98	3.44	7.35
16	2.29 ± 0.06	1.25 ± 0.06	193.13 ± 5.12	2.01 ± 0.14	2.75	4.69	2.65	7.11
17	1.59 ± 0.08	0.86 ± 0.11	184.07 ± 9.59	1.81 ± 0.01	5.20	13.06	5.21	0.39
18	4.10 ± 0.08	1.68 ± 0.11	223.82 ± 2.45	2.20 ± 0.09	2.04	6.45	1.09	4.24
19	2.83 ± 0.20	1.26 ± 0.11	201.78 ± 9.48	1.89 ± 0.22	7.16	8.61	4.70	11.67
20	2.78 ± 0.35	0.87 ± 0.15	204.80 ± 2.12	2.31 ± 0.28	12.65	17.70	1.04	12.12
21	3.22 ± 0.67	0.77 ± 0.07	195.44 ± 5.02	1.97 ± 0.04	20.91	8.57	2.57	1.91
22	1.72 ± 0.14	1.33 ± 0.03	152.10 ± 2.14	1.73 ± 0.13	8.33	2.01	1.41	7.36
23	1.73 ± 0.02	0.65 ± 0.12	157.93 ± 4.27	1.79 ± 0.03	1.04	18.85	2.70	1.79
24	2.15 ± 0.20	0.83 ± 0.11	213.88 ± 3.55	2.12 ± 0.04	9.29	13.35	1.66	1.99
25	0.53 ± 0.03	1.01 ± 0.28	114.45 ± 39.60	1.43 ± 0.43	6.62	27.72	34.60	30.02
26	2.56 ± 0.17	1.79 ± 0.34	196.23 ± 2.86	1.99 ± 0.10	6.73	19.18	1.46	5.07
27	1.88 ± 0.25	0.96 ± 0.08	203.79 ± 2.77	2.13 ± 0.08	13.14	8.17	1.36	3.78
28	4.39 ± 0.15	1.79 ± 0.11	216.32 ± 7.47	1.88 ± 0.07	3.35	6.38	3.45	3.52
29	3.07 ± 0.28	2.25 ± 0.14	174.04 ± 4.57	1.79 ± 0.11	9.03	6.40	2.63	6.02
30	3.51 ± 0.22	1.37 ± 0.16	178.60 ± 6.09	1.73 ± 0.13	6.36	11.48	3.41	7.28
31	2.93 ± 0.12	0.66 ± 0.07	160.22 ± 2.28	1.41 ± 0.17	4.15	11.13	1.42	12.27
32	1.98 ± 0.20	1.75 ± 0.21	191.41 ± 7.90	2.17 ± 0.31	10.17	12.13	4.13	14.22
33	0.70 ± 0.11	1.11 ± 0.08	119.08 ± 4.32	1.29 ± 0.03	15.51	7.03	3.62	2.64
34	2.26 ± 0.41	0.45 ± 0.00	156.43 ± 1.80	1.33 ± 0.04	18.36	0.56	1.15	3.26
35	2.23 ± 0.48	1.58 ± 0.16	203.53 ± 5.74	1.96 ± 0.24	21.50	9.91	2.82	12.24
36	1.56 ± 0.22	1.07 ± 0.32	97.56 ± 8.15	1.58 ± 0.03	14.41	30.46	8.36	2.11
37	3.28 ± 0.07	0.62 ± 0.02	157.49 ± 5.24	1.20 ± 0.12	2.22	3.61	3.33	10.15
38	4.15 ± 0.04	0.78 ± 0.07	185.36 ± 11.25	1.50 ± 0.04	1.03	9.21	6.07	2.41
39	2.11 ± 0.19	1.82 ± 0.21	164.57 ± 23.70	1.96 ± 0.74	9.01	11.51	14.40	37.91
40	1.47 ± 0.13	1.96 ± 0.06	169.36 ± 7.54	1.79 ± 0.10	8.67	3.24	4.45	5.86
41	2.47 ± 0.43	0.71 ± 0.09	158.31 ± 14.51	1.46 ± 0.32	17.35	12.85	9.16	21.66
42	4.20 ± 0.07	2.22 ± 0.65	170.70 ± 2.42	1.90 ± 0.02	1.68	29.47	1.42	1.20
43	5.90 ± 0.22	1.56 ± 0.37	225.78 ± 5.51	1.99 ± 0.05	3.69	23.81	2.44	2.76
44	1.98 ± 0.19	0.70 ± 0.02	157.50 ± 4.63	1.59 ± 0.05	9.69	2.29	2.94	2.99
45	2.58 ± 0.11	ND ^3^	216.02 ± 3.06	1.98 ± 1.12	4.47	-	1.42	56.55

^1^ expressed as mean ± SD; ^2^ γ-Τ is co-eluted with β-tocotrienol; ^3^ ND = Not Detected.

**Table 4 molecules-25-01614-t004:** Limits (min–max) of tocopherol content (mg/kg dried sample) for different vegetable oils. according to the Codex Alimentarius Commission on Fats and Oils Standard.

Oils	α-T	β-T	γ-T	δ-T	Total
almond	20–545	ND ^1^–10	5–104	ND–5	20–600
hazelnut	100–420	6–12	18–194	ND–10	200–600
walnut	ND–170	ND–110	120–400	ND–60	309–455
pistachio	10–330	ND	0–100	ND–50	100–600
flax/linseed	2–20	ND	100–712	3–14	150–905
avocado	50–450	ND	10–20	ND–10	50–450

^1^ ND = Not Detected.

**Table 5 molecules-25-01614-t005:** Peak correspondence of the pistachio oil FTIR spectra.

Wavenumber (cm^−1^)	Function Group	Abbreviations	Reference
3007	C-H symmetric stretching vibration of -CH_3_	v_s_(CH_3_)	[24,25,26]
2956	C-H asymmetric stretching vibration of -CH_3_	v_as_(CH_3_)	[27]
2922	C-H asymmetric stretching vibration of -CH_2_-	v_as_(CH_2_)	[24,25,26,28,29,30]
2853	C-H symmetric stretching vibration of -CH_2_-	v_s_(CH_2_)	[24,26,27,28,30]
1744	C=O stretching vibration	v(C=O)	[24,25,28,29]
1654	>C=C< cis-olefinic stretching vibration	v(C=C)	[24]
1461	C-H in-plane bending vibration of -CH_2_- (scissoring)	δ_s_(CH_2_)	[24,26,30,31]
1374	C-H symmetric bending vibration of -CH_3_	δ(CH_3_)	[25,27,28,30]
1345, 1313	-CH_2_- out-of-plane bending vibration (wagging)	ω(CH_2_)	[27]
1236, 1160, 1117	C-O asymmetric stretching vibration	ν_as_(C-O)	[25,27,28,30,31]
1095, 1029	in-phase-C-C stretching vibration	ν(C-C)	[27,30]
965	C-H in-plane bending vibration (scissoring)	δ_s_(C=C=C)	[27,28]
911, 857	-CH_2_- plane vibration	γ(CH_2_)	[27,28]
722	-CH=CH- cis-stretching vibration	v(C=C)	[24,28,29]

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
