# Peer review of "Study of the Quality Parameters and the Antioxidant Capacity for the FTIR-Chemometric Differentiation of Pistacia Vera Oils"

_molecules, 2020, doi:10.3390/molecules25071614_

Round 1

Reviewer 1 Report

The work is interesting and showed good results, however it does not present a clear order: I think that the abstract, introduction, material and methods must first be presented, then results and discussion as well as conclusion. This work after the introduction goes back to the result and discussion and  come back to material and Methods than go later the conclusion.

In conclusion, the authors conclude and do not suggest, in this topic the authors can summarize the results they found incisively. For example, the FTIR method is suitable for evaluate  the quality of pistachio oils.

Author Response

The work is interesting and showed good results, however it does not present a clear order: I think that the abstract, introduction, material and methods must first be presented, then results and discussion as well as conclusion. This work after the introduction goes back to the result and discussion and come back to material and Methods than go later the conclusion.

The template as provided by the journal “Molecules” was considered and followed.

In conclusion, the authors conclude and do not suggest, in this topic the authors can summarize the results they found incisively. For example, the FTIR method is suitable for evaluate the quality of pistachio oils.

The part of conclusions has been revised emphasizing to the obtained results and the utility of the method.

Reviewer 2 Report

I have only a quetion:

"A total of 45 pistachio samples of the greek variety ‘Aegina’ (weighted 1.5 kg each) were provided by
tachio farmers from eight different regions of Greece (Aegina, Megara, Phthiotis, Evia, Volos, Trizina,)".. was it = 45 x 1,5 kg = 67.5 kg in total??

Author Response

I have only a question: "A total of 45 pistachio samples of the greek variety ‘Aegina’ (weighted 1.5 kg each) were provided by pistachio farmers from eight different regions of Greece (Aegina, Megara, Phthiotis, Evia, Volos, Trizina,)".. was it = 45 x 1,5 kg = 67.5 kg in total?? Review Report Form.

The total quantity of each purchased pistachio sample was 1.5 kg which was stored and used for other experimental objectives. Hence this information is redundant and can be misleading for the readers was deleted (Line 303).

Reviewer 3 Report

The researcher demonstrates through their interesting work that FTIR spectroscopy could be used to classify pistachio oil samples according to the differences among quality parameters, especially the antioxidant capacity and the R-value.  This is a very good output of the research because the developed method is more sustainable for the environment and much cheaper. The results most probably can be extended to other type of oils, too.

Author Response

The researcher demonstrates through their interesting work that FTIR spectroscopy could be used to classify pistachio oil samples according to the differences among quality parameters, especially the antioxidant capacity and the R-value. This is a very good output of the research because the developed method is more sustainable for the environment and much cheaper. The results most probably can be extended to other type of oils, too.

Further experimental work is definitely required in order to develop a discrimination tool using FTIR spectroscopy combined with chemometrics applied in other types of oil. Additionally, the use of the FTIR method as a tool for the mitigation of food fraud incidences among plant-derived oils can be a future challenge.